# Tree-to-tree Neural Networks for Program Translation

**Xinyun Chen**
UC Berkeley
xinyun.chen@berkeley.edu

**Chang Liu**
UC Berkeley
liuchang2005acm@gmail.com

**Dawn Song**
UC Berkeley
dawnsong@cs.berkeley.edu

## Abstract

Program translation is an important tool to migrate legacy code in one language into an ecosystem built in a different language. In this work, we are the first to employ deep neural networks toward tackling this problem. We observe that program translation is a modular procedure, in which a sub-tree of the source tree is translated into the corresponding target sub-tree at each step. To capture this intuition, we design a tree-to-tree neural network to translate a source tree into a target one. Meanwhile, we develop an attention mechanism for the tree-to-tree model, so that when the decoder expands one non-terminal in the target tree, the attention mechanism locates the corresponding sub-tree in the source tree to guide the expansion of the decoder. We evaluate the program translation capability of our tree-to-tree model against several state-of-the-art approaches. Compared against other neural translation models, we observe that our approach is consistently better than the baselines with a margin of up to 15 points. Further, our approach can improve the previous state-of-the-art program translation approaches by a margin of 20 points on the translation of real-world projects.

## 1  Introduction

Programs are the main tool for building computer applications, the IT industry, and the digital world. Various programming languages have been invented to facilitate programmers to develop programs for different applications. At the same time, the variety of different programming languages also introduces a burden when programmers want to combine programs written in different languages together. Therefore, there is a tremendous need to enable program translation between different programming languages.

Nowadays, to translate programs between different programming languages, typically programmers would manually investigate the correspondence between the grammars of the two languages, then develop a rule-based translator. However, this process can be inefficient and error-prone. In this work, we make the first attempt to examine whether we can leverage deep neural networks to build a program translator automatically.

Intuitively, the program translation problem in its format is similar to a natural language translation problem. Some previous work propose to adapt phrase-based statistical machine translation (SMT) for code migration [21, 16, 22]. Recently, neural network approaches, such as sequence-to-sequence-based models, have achieved the state-of-the-art performance on machine translation [4, 9, 13, 14, 30]. In this work, we study neural machine translation methods to handle the program translation problem. However, a big challenge making a sequence-to-sequence-based model ineffective is that, unlike

natural languages, programming languages have rigorous grammars and are not tolerant to typos and grammatical mistakes. It has been demonstrated that it is very hard for an RNN-based sequence generator to generate syntactically correct programs when the lengths grow large [17].

In this work, we observe that the main issue of an RNN that makes it hard to produce syntactically correct programs is that it entangles two sub-tasks together: (1) learning the grammar; and (2) aligning the sequence with the grammar. When these two tasks can be handled separately, the performance can typically boost. For example, Dong et al. employ a tree-based decoder to separate the two tasks [11]. In particular, the decoder in [11] leverages the tree structural information to (1) generate the nodes at the same depth of the parse tree using an LSTM decoder; and (2) expand a non-terminal and generate its children in the parse tree. Such an approach has been demonstrated to achieve the state-of-the-art results on several semantic parsing tasks.

Inspired by this observation, we hypothesize that the structural information of both source and target parse trees can be leveraged to enable such a separation. Inspired by this intuition, we propose tree-to-tree neural networks to combine both a tree encoder and a tree decoder. In particular, we observe that in the program translation problem, both source and target programs have their parse trees. In addition, a cross-language compiler typically follows a modular procedure to translate the individual sub-components in the source tree into their corresponding target ones, and then compose them to form the final target tree. Therefore, we design the workflow of a tree-to-tree neural network to align with this procedure: when the decoder expands a non-terminal, it locates the corresponding sub-tree in the source tree using an attention mechanism, and uses the information of the sub-tree to guide the non-terminal expansion. In particular, a tree encoder is helpful in this scenario, since it can aggregate all information of a sub-tree to the embedding of its root, so that the embedding can be used to guide the non-terminal expansion of the target tree.

We follow the above intuition to design the tree-to-tree translation model. Some existing work [28, 18] propose tree-based autoencoder architectures. However, in these models, the decoder can only access to a single hidden vector representing the source tree, thus they are not performant on the translation task. In our evaluation, we demonstrate that without an attention mechanism, the translation performance is $0\%$ in most cases, while using an attention mechanism could boost the performance to $> 90\%$. Another work [6] proposes a tree-based attentional encoder-decoder architecture for natural language translation, but their model performs even worse than the attentional sequence-to-sequence baseline model. One main reason is that their attention mechanism calculates the attention weights of each node independently, which does not well capture the hierarchical structure of the parse trees. In our work, we design a *parent attention feeding* mechanism that formulates the dependence of attention maps between different nodes, and show that this attention mechanism further improves the performance of our tree-to-tree model considerably, especially when the size of the parse trees grows large (i.e., $20\% - 30\%$ performance gain). To the best of our knowledge, this is the first successful demonstration of tree-to-tree neural network architecture proposed for translation tasks in the literature.

To test our hypothesis, we develop two novel program translation tasks, and employ a Java to C# benchmark used by existing program translation works [22, 21]. First, we compare our approach against several neural network approaches on our proposed two tasks. Experimental results demonstrate that our tree-to-tree model outperforms other state-of-the-art neural networks on the program translation tasks, and yields a margin of up to $5\%$ on the token accuracy and up to $15\%$ on the program accuracy. Further, we compare our approach with previous program translation approaches on the Java to C# benchmark, and the results show that our tree-to-tree model outperforms previous state-of-the-art by a large margin of $20\%$ on program accuracy. These results demonstrate that our tree-to-tree model is promising toward tackling the program translation problem. Meanwhile, we believe that our proposed tree-to-tree neural network could also be adapted to other tree-to-tree tasks, and we consider it as future work.

## 2   Program Translation Problem

In this work, we consider the problem of translating a program in one language into another. One approach is to model the problem as a machine translation problem between two languages, and thus numerous neural machine translation approaches can be applied.

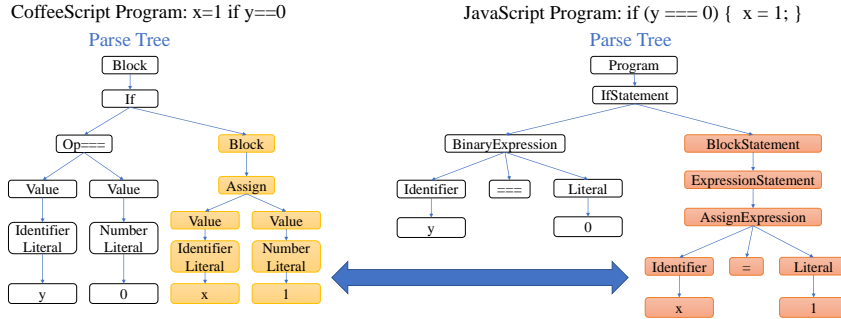

Figure 1: Translating a CoffeeScript program into JavaScript. The sub-component in the CoffeeScript program and its corresponding translation in JavaScript are highlighted.

For the program translation problem, however, a unique property is that each input program unambiguously corresponds to a unique parse tree. Thus, rather than modeling the input program as a sequence of tokens, we can consider the problem as translating a source tree into a target tree. Note that most modern programming languages are accompanied with a well-developed parser, so we can assume that the parse trees of both the source and the target programs can be easily obtained.

The main challenge of the problem in our consideration is that the cross-compiler for translating programs typically does not exist. Therefore, even if we assume the existence of parsers for both the source and the target languages, the translation problem itself is still non-trivial. We formally define the problem as follows.

**Definition 1** (Program translation). *Given two programming languages $\mathcal{L}_s$ and $\mathcal{L}_t$, each being a set of instances $(p_k, T_k)$, where $p_k$ is a program, and $T_k$ is its corresponding parse tree. We assume that there exists a translation oracle $\pi$, which maps instances in $\mathcal{L}_s$ to instances in $\mathcal{L}_t$. Given a dataset of instance pairs $(i_s, i_t)$ such that $i_s \in \mathcal{L}_s, i_t \in \mathcal{L}_t$ and $\pi(i_s) = i_t$, our problem is to learn a function $F$ that maps each $i_s \in \mathcal{L}_s$ into $i_t = \pi(i_s)$.*

In this work, we focus on the problem setting that we have a set of paired source and target programs to learn the translator. Note that all existing program translation works [16, 22, 21] also study the problem under such an assumption. When such an alignment is lacking, the program translation problem is more challenging. Several techniques for NMT have been proposed to handle this issue, such as dual learning [14], which have the potential to be extended for the program translation task. We leave these more challenging problem setups as future work.

## 3 Tree-to-tree Neural Network

In this section, we present our design of the tree-to-tree neural network. We first motivate the design, and then present the details.

### 3.1 Program Translation as a Tree-to-tree Translation Problem

Figure 1 presents an example of translation from CoffeeScript to JavaScript. We observe that an interesting property of the program translation problem is that the translation process can be modular. The figure highlights a sub-component in the source tree corresponding to `x=1` and its translation in the target tree corresponding to `x=1;`. This correspondence is independent of other parts of the program. Consider when the program grows longer and this statement may repetitively occur multiple times, it may be hard for a sequence-to-sequence model to capture the correspondence based on only token sequences without structural information. Thus, such a correspondence makes it a natural solution to locate the referenced sub-tree in the source tree when expanding a non-terminal in the target tree into a sub-tree.

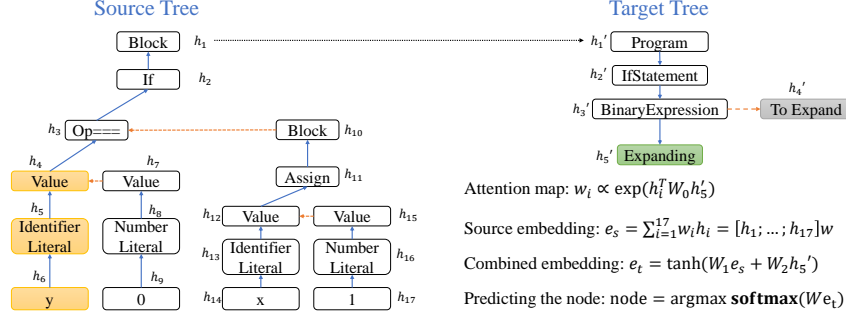

Figure 2: Tree-to-tree workflow: The arrows indicate the computation flow. Blue solid arrows indicate the flow from/to the left child, while orange dashed arrows are for the right child. The black dotted arrow from the source tree root to the target tree root indicates that the LSTM state is copied. The green box denotes the expanding node, and the grey one denotes the node to be expanded in the queue. The sub-tree of the source tree corresponding to the expanding node is highlighted in yellow. The right corner lists the formulas to predict the value of the expanding node.

## 3.2 Tree-to-tree Neural Network

Inspired by the above motivation, we design the tree-to-tree neural network, which follows an encoder-decoder framework to encode the source tree into an embedding, and decode the embedding into the target tree. To capture the intuition of the modular translation process, the decoder employs an attention mechanism to locate the corresponding source sub-tree when expanding the non-terminal. We illustrate the workflow of a tree-to-tree model in Figure 2, and present each component of the model below.

**Converting a tree into a binary one.** Note that the source and target trees may contain multiple branches. Although we can design tree-encoders and tree-decoders to handle trees with arbitrary number of branches, we observe that encoder and decoder for binary trees can be more effective. Thus, the first step is to convert both the source tree and the target tree into a binary tree. To this end, we employ the Left-Child Right-Sibling representation for this conversion.

**Binary tree encoder.** The encoder employs a Tree-LSTM [29] to compute embeddings for both the entire source tree and each of its sub-tree. In particular, consider a node $N$ with the value $t_s$ in its one-hot encoding representation, and it has two children $N_L$ and $N_R$, which are its left child and right child respectively. The encoder recursively computes the embedding for $N$ from the bottom up.

Assume that the left child and the right child maintain the LSTM state $(h_L, c_L)$ and $(h_R, c_R)$ respectively, and the embedding of $t_s$ is $x$. Then the LSTM state $(h, c)$ of $N$ is computed as

$$(h, c) = \text{LSTM}(([h_L; h_R], [c_L; c_R]), x) \tag{1}$$

where $[a; b]$ denotes the concatenation of $a$ and $b$. Note that a node may lack one or both of its children. In this case, the encoder sets the LSTM state of the missing child to be zero.

**Binary tree decoder.** The decoder generates the target tree starting from a single root node. The decoder first copies the LSTM state $(h, c)$ of the root of the source tree, and attaches it to the root node of the target tree. Then the decoder maintains a queue of all nodes to be expanded, and recursively expands each of them. In each iteration, the decoder pops one node from the queue, and expands it. In the following, we call the node being expanded *the expanding node*.

First, the decoder will predict the value of expanding node. To this end, the decoder computes the embedding $e_t$ of the expanding node $N$, and then feeds it into a softmax regression network for prediction:

$$\text{t}_t = \textbf{argmax}\ \textbf{softmax}(We_t) \tag{2}$$

Here, $W$ is a trainable matrix of size $V_t \times d$, where $V_t$ is the vocabulary size of the outputs and $d$ is the embedding dimension. Note that $e_t$ is computed using the attention mechanism, which we will explain later.

The value of each node $t_t$ is a non-terminal, a terminal, or a special $\langle \text{EOS} \rangle$ token. If $t_t = \langle \text{EOS} \rangle$, then the decoder finishes expanding this node. Otherwise, the decoder generates one new node as the left child and another new node as the right child of the expanding one. Assume that $(h', c')$, $(h'', c'')$ are the LSTM states of its left child and right child respectively, then they are computed as:

$$(h', c') = \text{LSTM}_L((h, c), Bt_t) \tag{3}$$

$$(h'', c'') = \text{LSTM}_R((h, c), Bt_t) \tag{4}$$

Here, $B$ is a trainable word embedding matrix of size $d \times V_t$. Note that the generation of the left child and right child use two different sets of parameters for $\text{LSTM}_L$ and $\text{LSTM}_R$ respectively. These new children are pushed into the queue of all nodes to be expanded. When the queue is empty, the target tree generation process terminates.

Notice that although the sets of terminal and non-terminal are disjoint, it is necessary to include the $\langle \text{EOS} \rangle$ token for the following reasons. First, due to the left-child-right-sibling encoding, although a terminal does not have a child, since it could have a right child representing its sibling in the original tree, $\langle \text{EOS} \rangle$ is still needed for predicting the right branch. Meanwhile, we combine the terminal and non-terminal sets into a single vocabulary $V_t$ for the decoder, and do not incorporate the knowledge of grammar rules into the model, thus the model needs to infer whether a predicted token is a terminal or a non-terminal itself. In our evaluation, we find that a well-trained model never generates a left child for a terminal, which indicates that the model can learn to distinguish between terminals and non-terminals correctly.

**Attention mechanism to locate the source sub-tree.** Now we consider how to compute $e_t$. One straightforward approach is to compute $e_t$ as $h$, which is the hidden state attached to the expanding node. However, in doing so, the embedding will soon forget the information about the source tree when generating deep nodes in the target tree, and thus the model yields a very poor performance.

To make better use of the information of the source tree, our tree-to-tree model employs an attention mechanism to locate the source sub-tree corresponding to the sub-tree rooted at the expanding node. Specifically, we compute the following probability:

$$P(N_s \text{ is the source sub-tree corresponding to } N_t | N_t)$$

where $N_t$ is the expanding node. We denote this probability as $P(N_s|N_t)$, and we compute it as

$$P(N_s|N_t) \propto \mathbf{exp}(h_s^T W_0 h_t) \tag{5}$$

where $W_0$ is a trainable matrix of size $d \times d$.

To leverage the information from the source tree, we compute the expectation of the hidden state value across all $N_s$ conditioned on $N_t$, i.e.,

$$e_s = \text{E}[h_{N_s}|N_t] = \sum_{N_s} h_{N_s} \cdot P(N_s|N_t) \tag{6}$$

This embedding can then be combined with $h$, the hidden state of the expanding node, to compute $e_t$ as follows:

$$e_t = \mathbf{tanh}(W_1 e_s + W_2 h) \tag{7}$$

where $W_1$, $W_2$ are trainable matrices of size $d \times d$ respectively.

**Parent attention feeding.** In the above approach, the attention vectors $e_t$ are computed independently to each other, since once $e_t$ is used for predicting the node value $t_t$, $e_t$ is no longer used for further predictions. However, intuitively, the attention decisions for the prediction of each node should be related to each other. For example, for a non-terminal node $N_t$ in the target tree, suppose that it is related to $N_s$ in the source tree, then it is very likely that the attention weights of its children should focus on the descendants of $N_s$. Therefore, when predicting the attention vector of a node, the model should leverage the attention information of its parent as well.

Following this intuition, we propose a *parent attention feeding* mechanism, so that the attention vector of the expanding node is taken into account when predicting the attention vectors of its children. Formally, besides the embedding of the node value $t_t$, we modify the inputs to $\text{LSTM}_L$ and $\text{LSTM}_R$ of the decoder in Equations (3) and (4) as below:

$$(h', c') = \text{LSTM}_L((h, c), [Bt_t; e_t]) \tag{8}$$

$$(h'', c'') = \text{LSTM}_R((h, c), [Bt_t; e_t]) \tag{9}$$

Notice that these formulas in their formats coincide with the input-feeding method for sequential neural networks [20], but their meanings are different. For sequential models, the input attention vector belongs to the previous token, while here it belongs to the parent node. In our evaluation, we will show that such a parent attention feeding mechanism significantly improves the performance of our tree-to-tree model.

## 4 Evaluation

In this section, we evaluate our tree-to-tree neural network with several baseline approaches on the program translation task. To do so, we first describe three benchmark datasets in Section 4.1 for evaluating different aspects; then we evaluate our tree-to-tree model against several baseline approaches, including the state-of-the-art neural network approaches and program translation approaches.

### 4.1 Datasets

To evaluate different approaches for the program translation problem, we employ three tasks: (1) a synthetic translation task from an imperative language to a functional language; (2) translation between CoffeeScript and JavaScript, which are both full-fledged languages; and (3) translation of real-world projects from Java to C#, which has been used as a benchmark in the literature. Due to the space limit, we present the translation tasks of real-world programming languages (i.e., task (2) and (3)) below, and we discuss the synthetic task in the supplementary material.

For the CoffeeScript-JavaScript task, CoffeeScript employs a Python-style succinct syntax, while JavaScript employs a C-style verbose syntax. To control the program lengths of the training and test data, we develop a pCFG-based program generator and a subset of the core CoffeeScript grammar. We also limit the set of variables and literals to restrict the vocabulary size. We utilize the CoffeeScript compiler to generate the corresponding ground truth JavaScript programs. The grammar used to generate the programs in our experiments can be found in the supplementary material. In doing so, we obtain a set of CoffeeScript-JavaScript pairs, and thus we can build a CoffeeScript-to-JavaScript dataset, and a JavaScript-to-CoffeeScript dataset by exchanging the source and the target. To build the dataset, we randomly generate 100,000 pairs of source and target programs for training, 10,000 pairs as the development set, and 10,000 pairs for testing. We guarantee that there is no overlap among training, development and test sets, and all samples are unique in the dataset. More statistics of the dataset can be found in the supplementary material.

For the evaluation on Java to C#, we tried to contact the authors of [22] for their dataset, but our emails were not responded. Thus, we employ the same approach as in [22] to crawl several open-source projects, which have both a Java and a C# implementation. Same as in [22], we pair the methods in Java and C# based on their file names and method names. The statistics of the dataset is summarized in the supplementary material. Due to the change of the versions of these projects, the concrete dataset in our evaluation may differ from [22]. For each project, we apply ten-fold validation on matched method pairs, as in [22].

### 4.2 Metrics

The main metric evaluated in our evaluation is the *program accuracy*, which is the percentage of the predicted target programs that are exactly the same as the ground truth in the dataset. Note that the program accuracy is an underestimation of the true accuracy based on semantic equivalence, and this metric has been used in [22]. This metric is more meaningful than other previously proposed metrics, such as syntax-correctness and dependency-graph-accuracy, which are not directly comparable to semantic equivalence. We also measure another metric called *token accuracy*, and we defer the details to the supplementary material.

| | Tree2tree | | | Seq2seq | | | | Seq2tree | | Tree2seq | |
|---|---|---|---|---|---|---|---|---|---|---|---|
| | T→T | T→T (-PF) | T→T (-Attn) | P→P | P→T | T→P | T→T | P→T | T→T | T→P | T→T |
| CoffeeScript to JavaScript translation | | | | | | | | | | | |
| CJ-AS | **99.57%** | 98.80% | 0.09% | 90.51% | 79.82% | 92.73% | 89.13% | 86.52% | 88.50% | 96.96% | 92.18% |
| CJ-BS | **99.75%** | 99.67% | 0% | 97.44% | 16.26% | 98.05% | 93.89% | 91.97% | 88.22% | 96.83% | 78.77% |
| CJ-AL | **97.15%** | 71.52% | 0% | 21.04% | 0% | 0% | 0% | 80.82% | 78.60% | 82.55% | 46.94% |
| CJ-BL | **95.60%** | 78.61% | 0% | 19.26% | 9.98% | 25.35% | 42.08% | 76.12% | 76.21% | 83.61% | 26.83% |
| JavaScript to CoffeeScript translation | | | | | | | | | | | |
| JC-AS | **87.75%** | 85.11% | 0.09% | 83.07% | 86.13% | 73.88% | 86.31% | 86.86% | 86.99% | 71.61% | 86.53% |
| JC-BS | **86.37%** | 80.35% | 0% | 80.49% | 85.94% | 69.77% | 85.28% | 85.06% | 84.25% | 66.82% | 85.31% |
| JC-AL | **78.59%** | 54.93% | 0% | 77.10% | 77.30% | 65.52% | 75.70% | 77.11% | 77.59% | 60.75% | 75.75% |
| JC-BL | **75.62%** | 44.40% | 0% | 73.14% | 73.96% | 61.92% | 74.51% | 74.34% | 71.56% | 57.09% | 73.86% |

Table 1: Program accuracy for the translation between CoffeeScript and JavaScript.

## 4.3 Model Details

We evaluate our tree-to-tree model against a sequence-to-sequence model [4, 31], a sequence-to-tree model [11], and a tree-to-sequence model [13]. Note that for a sequence-to-sequence model, there can be four variants to handle different input-output formats. For example, given a program, we can simply tokenize it into a sequence of tokens. We call this format as *raw program*, denoted as P. We can also use the parser to parse the program into a parse tree, and then serialize the parse tree as a sequence of tokens. Our serialization of a tree follows its depth-first traversal order, which is the same as [31]. We call this format as *parse tree*, denoted as T. For both input and output formats, we can choose either P or T. For a sequence-to-tree model, we have two variants based on its input format being either P or T; note that the sequence-to-tree model generates a tree as output, and thus requires its output format to be T (unserialized). Similarly, the tree-to-sequence model has two variants, and our tree-to-tree only has one form. Therefore, we have 9 different models in our evaluation.

The hyper-parameters used in different models can be found in the supplementary material. The baseline models have employed their own input-feeding or parent-feeding method that is analogous to our parent attention feeding mechanism.

## 4.4 Results on the CoffeeScript-JavaScript Task

For the CoffeeScript-JavaScript task, we create several datasets named as XY-ZW: X and Y (C or J) indicate the source and target languages respectively; Z (A or B) indicates the vocabulary; and W (S or L) indicates the program length. In particular, vocabulary A uses $\{x, y\}$ as variable names and $\{0, 1\}$ as literals; vocabulary B uses all alphabetical characters as variable names, and all single digits as literals. S means that the CoffeeScript programs has 10 tokens on average; and L for 20.

The program accuracy results are presented in Table 1. We can observe that our tree2tree model outperforms all baseline models on all datasets. Especially, on the dataset with longer programs, the program accuracy significantly outperforms all seq2seq models by a large margin, i.e., up to $75\%$. Its margin over a seq2tree model can also reach around 20 points. These results demonstrate that tree2tree model is more capable of learning the correspondence between the source and the target programs; in particular, it is significantly better than other baselines at handling longer inputs.

Meanwhile, we perform an ablation study to compare the full tree2tree model with (1) tree2tree without parent attention feeding (T→T (-PF)) and (2) tree2tree without attention (T→T (-Attn)). We observe that the full tree2tree model significantly outperforms the other alternatives. In particular, on JC-BL, the full tree2tree's program accuracy is 30 points higher than the tree2tree model without parent attention feeding.

More importantly, we observe that the program accuracy of tree2tree model without the attention mechanism is nearly $0\%$. Note that such a model is similar to a tree-to-tree autoencoder architecture. This result shows that our novel architecture can significantly outperform previous tree-to-tree-like architectures on the program translation task.

However, although our tree2tree model performs better than other baselines, it still could not achieve $100\%$ accuracy. After investigating into the prediction, we find that the main reason is because the translation may introduce temporary variables. Because such temporary variables appear very rarely in the training set, it could be hard for a neural network to infer correctly in these cases. Actually,

|          | Tree2tree | J2C# | 1pSMT | mppSMT |
|----------|-----------|------|-------|--------|
|          |           | Reported in [22] | | |
| Lucene   | **72.8%** | 21.5% | 21.6% | 40.0% |
| POI      | **72.2%** | 18.9% | 34.6% | 48.2% |
| Itext    | **67.5%** | 25.1% | 24.4% | 40.6% |
| JGit     | **68.7%** | 10.7% | 23.0% | 48.5% |
| JTS      | **68.2%** | 11.7% | 18.5% | 26.3% |
| Antlr    | 31.9% (**58.3%**) | 10.0% | 11.5% | 49.1% |

Table 2: Program accuracy on the Java to C# translation. In the parentheses, we present the program accuracy that can be achieved by increasing the training set.

the longer the programs are, the more temporary variables that the cross-compiler may introduce, which makes the prediction harder. We consider further improving the model to handle this problem as future work.

In addition, we observe that for the translation from JavaScript to CoffeeScript, the improvements of the tree2tree model over the baselines are much smaller than for CoffeeScript to JavaScript translation. We attribute this to the fact that the target programs are much shorter. For example, for a CoffeeScript program with 20 tokens, its corresponding JavaScript program may contain more than 300 tokens. Thus, the model needs to predict much fewer tokens for a CoffeeScript program than a JavaScript program, so that even seq2seq models can achieve a reasonably good accuracy. However, still, we can observe that our tree2tree model outperforms all baselines.

### 4.5 Results on Real-world Projects

We now compare our approach with three state-of-the-art program translation approaches, i.e., J2C# [15], 1pSMT [21], and mppSMT [22], on the real-world benchmark from Java to C#. Here, J2C# is a rule-based system, 1pSMT directly applies the phrase-based SMT on sequential programs, and mppSMT is a multi-phase phrase-based SMT approach that leverages both the raw programs and their parse trees.

The results are summarized in Table 2. For previous approaches, we report the results from [22]. We can observe that our tree2tree approach can significantly outperform the previous state-of-the-art on all projects except Antlr. The improvements range from $20.2\%$ to $41.9\%$.

On Antlr, the tree2tree model performs worse. We attribute this to the fact that Antlr contains too few data samples for training. We test our hypothesis by constructing another training and validation set from all other 5 projects, and test our model on the entire Antlr. We observe that our tree2tree model can achieve a test accuracy of $58.3\%$, which is 9 points higher than the state-of-the-art. Therefore, we conclude that our approach can significantly outperform previous program translation approaches when there are sufficient training data.

## 5 Related Work

**Statistical approaches for program translation.** Some recent work have applied statistical machine translation techniques to program translation [2, 16, 22, 21, 23, 24]. For example, several works propose to adapt phrase-based statistical machine translation models and leverage grammatical structures of programming languages for code migration [16, 22, 21]. In [23], Nguyen et al. propose to use Word2Vec representation for APIs in libraries used in different programming languages, then learn a transformation matrix for API mapping. On the contrary, our work is the first to employ deep learning techniques for program translation.

**Neural networks with tree structures.** Recently, various neural networks with tree structures have been proposed to employ the structural information of the data [11, 26, 25, 32, 3, 29, 34, 27, 13, 33, 28, 18, 6]. In these work, different tree-structured encoders are proposed for embedding the input data, and different tree-structured decoders are proposed for predicting the output trees. In particular, in [28, 18], they propose tree-structured autoencoders to learn vector representations of trees, and show better performance on tree reconstruction and other tasks such as sentiment analysis. Another work [6] proposes to use a tree-structured encoder-decoder architecture for natural language

translation, where both the encoder and the decoder are variants of the RNNG model [12]; however, the performance of their model is slightly worse than the sequence-to-sequence model with attention, which is mainly due to the fact that their attention mechanism can not condition the future attention weights on previously computed ones. In this work, we are the first to demonstrate a successful design of tree-to-tree neural network for translation tasks.

**Neural networks for parsing.** Other work study using neural networks to generate parse trees from input-output examples [11, 31, 1, 26, 32, 3, 12, 8, 7]. In [11], Dong et al. propose a seq2tree model that allows the decoder RNN to generate the output tree recursively in a top-down fashion. This approach achieves the state-of-the-art results on several semantic parsing tasks. Some other work incorporate the knowledge of the grammar into the architecture design [32, 26] to achieve better performance on specific tasks. However, these approaches are hard to generalize to other tasks. Again, none of them is designed for program translation or proposes a tree-to-tree architecture.

**Neural networks for code generation.** A recent line of research study using neural networks for code generation [5, 10, 25, 19, 26, 32]. In [19, 26, 32], they study generating code in a DSL from inputs in natural language or in another DSL. However, their designs require additional manual efforts to adapt to new DSLs in consideration. In our work, we consider the tree-to-tree model as a generic approach that can be applied to any grammar.

## 6 Conclusion and Future Work

In this work, we are the first to consider neural network approaches for the program translation problem, and are the first to demonstrate a successful design of tree-to-tree neural network combining both a tree-RNN encoder and a tree-RNN decoder for translation tasks. Extensive evaluation demonstrates that our tree-to-tree neural network outperforms several state-of-the-art models. This renders our tree-to-tree model as a promising tool toward tackling the program translation problem. In addition, we believe that our proposed tree-to-tree neural network has the potential to generalize to other tree-to-tree tasks, and we consider it as future work.

At the same time, we observe many challenges in program translation that existing techniques are not capable of handling. For example, the models are hard to generalize to programs longer than the training ones; it is unclear how to handle an infinite vocabulary set that may be employed in real-world applications; further, the training requires a dataset of aligned input-output pairs, which may be lacking in practice. We consider all these problems as important future work in the research agenda toward solving the program translation problem.

## Acknowledgement

We thank the anonymous reviewers for their valuable comments. This material is in part based upon work supported by the National Science Foundation under Grant No. TWC-1409915, Berkeley DeepDrive, and DARPA D3M under Grant No. FA8750-17-2-0091. Any opinions, findings, and conclusions or recommendations expressed in this material are those of the author(s) and do not necessarily reflect the views of the National Science Foundation.

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
