[Supplementary Material]

# Supplementary Material for "Tree-to-tree Neural Networks for Program Translation"

## A  Hyper-parameters of Neural Network Models

|  | Seq2seq | Seq2tree | Tree2seq | Tree2tree |
|---|---|---|---|---|
| Batch size | 100 | 20 | 100 | 100 |
| Number of RNN layers | 3 | 1 | 1 | 1 |
| Encoder RNN cell | LSTM | LSTM | Tree LSTM | Tree LSTM |
| Decoder RNN cell | LSTM | | | |
| Initial learning rate | 0.005 | | | |
| Learning rate decay schedule | Decay the learning rate by a factor of $0.8\times$ when the validation loss does not decrease for 500 mini-batches | | | |
| Hidden state size | 256 | | | |
| Embedding size | 256 | | | |
| Dropout rate | 0.5 | | | |
| Gradient clip threshold | 5.0 | | | |
| Weights initialization | Uniformly random from [-0.1, 0.1] | | | |

Table 1: Hyper-parameters chosen for each neural network model.

We present the hyper-parameters of different neural networks in Table 1. These hyper-parameters are chosen to achieve the best accuracy on the development set through a grid search.

## B  More Statistics of the Datasets

We present more detailed statistics of the datasets for the CoffeeScript-JavaScript task and the translation of real-world projects from Java to C# in Table 2 and 3 respectively.

## C  More Results on the CoffeeScript-JavaScript Task

Besides the program accuracy, we also measure the token accuracy of different approaches, which is the percentage of the tokens that are exactly the same as the ground truth. This metric is a finer-grained measurement of the correctness, thus provides some additional insights of the performance of different models.

Table 4 shows the token accuracy of different approaches for the translation between CoffeeScript and JavaScript.

## D  Grammar for the CoffeeScript-JavaScript Task

The grammar used to generate the CoffeeScript-JavaScript dataset, which is a subset of the core CoffeeScript grammar, is provided in Figure 1.

|  | CJ-(A/B)S | CJ-(A/B)L |
|---|---|---|
| Average input length (P) | 10 | 20 |
| Minimal output length (P) | 23 | 33 |
| Maximal output length (P) | 151 | 311 |
| Average output length (P) | 44 | 69 |
| Minimal input length (T) | 34 | 69 |
| Maximal input length (T) | 61 | 111 |
| Average input length (T) | 48 | 85 |
| Minimal output length (T) | 38 | 73 |
| Maximal output length (T) | 251 | 531 |
| Average output length (T) | 71 | 129 |

Table 2: Statistics of the datasets used for the CoffeeScript-JavaScript task.

| Project | # of matched methods |
|---|---|
| Lucene [5] | 5,516 |
| POI [6] | 3,153 |
| Itext [2] | 3,079 |
| JGit [3] | 2,780 |
| JTS [4] | 2,003 |
| Antlr [1] | 465 |
| Total | 16,996 |

Table 3: Statistics of the Java to C# dataset.

| | Tree2tree | | | Seq2seq | | | | Seq2tree | | Tree2seq | |
|---|---|---|---|---|---|---|---|---|---|---|---|
| | T→T | T→T (-PF) | T→T (-Attn) | P→P | P→T | T→P | T→T | P→T | T→T | T→P | T→T |
| CoffeeScript to JavaScript translation | | | | | | | | | | | |
| CJ-AS | **99.97%** | **99.97%** | 56.21% | 93.51% | 92.30% | 95.46% | 95.05% | 93.29% | 95.94% | 98.96% | 98.09% |
| CJ-BS | **99.98%** | **99.98%** | 47.54% | 99.08% | 87.51% | 99.11% | 96.14% | 98.31% | 98.09% | 99.27% | 98.10% |
| CJ-AL | **99.37%** | 98.16% | 32.99% | 85.84% | 25.65% | 19.13% | 36.18% | 95.64% | 94.74% | 94.18% | 84.71% |
| CJ-BL | **99.36%** | 99.27% | 31.80% | 80.22% | 63.49% | 87.27% | 79.85% | 94.09% | 94.64% | 93.85% | 78.07% |
| JavaScript to CoffeeScript translation | | | | | | | | | | | |
| JC-AS | **99.14%** | 98.81% | 65.42% | 88.44% | 96.27% | 88.46% | 98.34% | 98.20% | 99.06% | 86.93% | 98.36% |
| JC-BS | **98.84%** | 98.18% | 55.22% | 86.85% | 97.92% | 85.98% | 98.09% | 96.93% | **98.84%** | 84.81% | 97.94% |
| JC-AL | **96.95%** | 92.65% | 42.23% | 88.09% | 95.94% | 87.19% | 95.04% | 93.51% | 96.59% | 84.57% | 94.63% |
| JC-BL | **96.48%** | 92.49% | 39.89% | 87.31% | 94.12% | 85.70% | 96.24% | 94.79% | 96.33% | 83.03% | 94.68% |

Table 4: Token accuracy of different approaches for translation between CoffeeScript and JavaScript.

# E    Evaluation on the Synthetic Task

In the following, we discuss our synthetic translation task from an imperative language to a functional language.

## E.1   Evaluation Setup

For the synthetic task, we design an imperative source language and a functional target language. Such a design makes the source and target languages use different programming paradigms, so that the translation can be challenging. Figure 2 illustrates an example of the translation, which demonstrates that a for-loop is translated into a recursive function. We manually implement a translator, which is used to acquire the ground truth. The grammar specifications of the source language (FOR language) and the target language (LAMBDA language) are provided in Figure 3 and Figure 4 respectively. The python source code to implement the translator from a FOR program to a LAMBDA program is provided in Figure 5.

To build the dataset, similar to the CoffeeScript-JavaScript task, we randomly generate 100,000 pairs of source and target programs for training, 10,000 pairs as the development set, and 10,000 pairs for testing. We guarantee that there is no overlap among training, development and test sets, and all samples are unique in the dataset. More statistics of the dataset can be found in Table 6.

```
        <Expr>  ::=  <Var>
                 |   <Const>
                 |   <Expr> + <Var>
                 |   <Expr> + <Const>
                 |   <Expr> * <Var>
                 |   <Expr> * <Const>
                 |   <Expr> == <Var>
                 |   <Expr> == <Const>
      <Simple>  ::=  <Var> = <Expr>
                 |   <Expr>
     <IfShort>  ::=  <Simple> if <Expr>
                 |   <IfShort> if <Expr>
  <WhileShort>  ::=  <Simple> while <Expr>
                 |   <WhileShort> while <Expr>
<ShortStatement>  ::=  <Simple> | <IfShort> | <WhileShort>
   <Statement>  ::=  <ShortStatement>
                 |   if <Expr> <br> <indent+> <Block> <indent->
                 |   while <Expr> <br> <indent+> <Block> <indent->
                 |   if <Expr> <br> <indent+> <Block> <indent-> <br>
                     else <br> <indent+> <Block> <indent->
                 |   if <Expr> then <ShortStatement> else <ShortStatement>
       <Block>  ::=  <Statement>
                 |   <Block> <br> <Statement>
```

Figure 1: A subset of the CoffeeScript grammar used to generate the CoffeeScript-JavaScript dataset. Here, <br> denotes the newline character.

```
Source program              Target program
for i=1; i<10; i+1 do       letrec f i =
    if x>1 then                 if i<10 then
           y=1                      let _ = if x>1 then
    else                                 let y=1 in ()
        y=2                          else let y=2 in ()
endfor                           in f i+1
                                else ()
                            in f 1
```

Figure 2: An example of the translation for the synthetic task.

### E.2 Results on the Synthetic Task

We create two datasets for the synthetic task: one with an average length of 20 (*SYN-S*) and the other with an average length of 50 (*SYN-L*). Here, the length of a program indicates the number of tokens in the source program.

We present the results in Table 5. Our observations are consistent with the results of the CoffeeScript-JavaScript task: our tree2tree model outperforms all baseline models; all models perform worse on longer inputs; both the attention and the parent attention feeding mechanisms boost the performance of our tree2tree model significantly.

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

| | Tree2tree | | | Seq2seq | | | | Seq2tree | | Tree2seq | |
|---|---|---|---|---|---|---|---|---|---|---|---|
| | T→T | T→T (-PF) | T→T (-Attn) | P→P | P→T | T→P | T→T | P→T | T→T | T→P | T→T |
| | Token accuracy | | | | | | | | | | |
| SYN-S | **99.99%** | 99.95% | 55.60% | 99.75% | 99.59% | 99.90% | 99.73% | 99.70% | 99.51% | 99.88% | 99.82% |
| SYN-L | **99.60%** | 96.68% | 34.48% | 68.31% | 45.28% | 67.37% | 35.01% | 96.95% | 97.41% | 97.08% | 95.88% |
| | Program accuracy | | | | | | | | | | |
| SYN-S | **99.76%** | 98.61% | 0% | 97.92% | 97.35% | 98.38% | 98.18% | 96.14% | 98.01% | 98.51% | 98.36% |
| SYN-L | **97.50%** | 57.42% | 0% | 12.19% | 0% | 9.19% | 0% | 67.34% | 68.11% | 91.35% | 87.84% |

Table 5: Token accuracy and program accuracy of different approaches for the synthetic task.

| | SYN-S | SYN-L |
|---|---|---|
| Average input length (P) | 20 | 50 |
| Minimal output length (P) | 22 | 46 |
| Maximal output length (P) | 44 | 96 |
| Average output length (P) | 30 | 71 |
| Minimal input length (T) | 40 | 100 |
| Maximal input length (T) | 56 | 134 |
| Average input length (T) | 49 | 111 |
| Minimal output length (T) | 41 | 90 |
| Maximal output length (T) | 82 | 177 |
| Average output length (T) | 55 | 133 |

Table 6: Statistics of the datasets used for the synthetic task.

[5] Lucene. Lucene. http://lucene.apache.org/, 2018.

[6] POI. Poi. http://poi.apache.org/, 2018.

```
    <Expr>  ::=  <Var>
              |  <Const>
              |  <Expr> + <Var>
              |  <Expr> + <Const>
              |  <Expr> − <Var>
              |  <Expr> − <Const>
     <Cmp>  ::=  <Expr> == <Expr>
              |  <Expr> > <Expr>
              |  <Expr> < <Expr>
  <Assign>  ::=  <Var> = <Expr>
      <If>  ::=  if <Cmp> then <statement>
                 else <statement> endif
     <For>  ::=  for <Var> = <Expr> ;
                 <Cmp> ; <Expr> do
                 <Statement> endfor
  <Single>  ::=  <Assign> | <If> | <For>
     <Seq>  ::=  <Single> ; <Single>
              |  <Seq> ; <Single>
<Statement> ::=  <Seq> | <Single>
```

Figure 3: Grammar for the source language FOR in the synthetic task.

```
   <Unit>  ::=  ()
    <App>  ::=  <Var> <Expr>
             |  <App> <Expr>
   <Expr>  ::=  <Var>
             |  <Expr> + <Var>
             |  <Expr> − <Var>
    <Cmp>  ::=  <Expr> == <Expr>
             |  <Expr> > <Expr>
             |  <Expr> < <Expr>
   <Term>  ::=  <LetTerm> | <Expr> | <Unit>
             |  <IfTerm> | <App>
<LetTerm>  ::=  let <Var> = <Term> in <Term>
             |  letrec <Var> <Var> = <Term>
                in <Term>
 <IfTerm>  ::=  if <Cmp> then <Term>
                else <Term>
```

Figure 4: Grammar for the target language LAMBDA in the synthetic task.

```python
def translate_from_for(self, ast):
    if type(ast) == type([]):
        if ast[0] == '<SEQ>':
            t1 = self.translate_from_for(ast[1])
            t2 = self.translate_from_for(ast[2])
            if t1[0] == '<LET>' and t1[-1] == '<UNIT>':
                t1[-1] = t2
                return t1
            else:
                return ['<LET>', 'blank', t1, t2]
        elif ast[0] == '<IF>':
            cmp = ast[1]
            t1 = self.translate_from_for(ast[2])
            t2 = self.translate_from_for(ast[3])
            return ['<IF>', cmp, t1, t2]
        elif ast[0] == '<FOR>':
            var = self.translate_from_for(ast[1])
            init = self.translate_from_for(ast[2])
            cmp = self.translate_from_for(ast[3])
            inc = self.translate_from_for(ast[4])
            body = self.translate_from_for(ast[5])
            tb = ['<LET>', 'blank', body, ['<APP>', 'func', inc]]
            func_body = ['<IF>', cmp, tb, '<UNIT>']
            translate = ['<LETREC>', 'func', var, func_body, ['<APP>', '
    func', inc]]
            return translate
        elif ast[0] == '<ASSIGN>':
            return ['<LET>', ast[1], ast[2], '<UNIT>']
        elif ast[0] == '<Expr>':
            return ast
        elif ast[0] == '<Op+>':
            return ast
        elif ast[0] == '<Op->':
            return ast
        elif ast[0] == '<CMP>':
            return ast
    else:
        return ast
```

Figure 5: The Python code to translate a FOR program into a LAMBDA program in the synthetic task.