[Reviews · NeurIPS 2018]

Reviewer 1



This work introduces an attentional tree-to-tree neural network for programming language translation, and evaluates it on two synthetic tasks and one real-world dataset. The results are significantly better than previous work in programming language translation, including rule-based and sequence-to-sequence approaches. But one central claim--that the authors "are the first to design the tree-to-tree neural network for translation tasks"--is inaccurate. I am familiar with a previous version of this paper presented at the 2018 ICLR Workshop, and with the comments from the anonymous ICLR reviewers available at OpenReview; I believe the authors did a good job of improving their paper in response to those reviews. Most importantly, reviewers asked for evaluation on real-world data; the authors then went to great lengths to replicate the evaluation methodology of Nguyen et al. (2015) despite receiving no response to emails they sent, resulting in a strong (though perhaps difficult to compare exactly) real world evaluation result. The model is described clearly and comprehensively; it would be relatively straightforward to reimplement from the paper alone. The two most important contributions of the model--the use of tree attention, and the use of parent-feeding--are both ablated in the results. As far as the primacy claim: the work indeed fills a significant void in the literature. But the first tree-to-tree neural network (also for translation, and also using tree attention) was Bradbury and Socher (2017), presented at an EMNLP 2017 workshop (http://aclweb.org/anthology/W17-4303), and it would be helpful to cite and compare to that work, although the results of that paper were relatively weak and it wasn't published at a conference.

Reviewer 2



Summary This paper proposes a neural network architecture for tree to tree transduction, with application to program translation. Binarized trees are encoded with a TreeLSTM, and a tree-structured decoder generates binary trees breadth-first, using an attention mechanism over input tree nodes, and attention feeding within the tree decoder. On synthetic and real-world program translation tasks, and across multiple datasets, the proposed model obtains substantially stronger performance than baseline sequential neural models and previous work on automatic program translation. Quality The paper proposes a general framework for neural tree to tree transduction, and the model obtains high performance on program translation tasks. Clarity The model and experimental setup are define clearly. If the terminal and non-terminal vocabularies are disjoint, is it necessary to generate end-of-sentence tokens explicitly, as they should always follow terminals? Originality Previous neural network architectures have been proposed for code generation (as cited in the paper), but have not been applied to program translation. The novelty of the proposed model lies in having both a tree-structured encoder and decoder. The proposed decoder generates trees breadth-first rather than depth-first, as it typical in many other architectures. To show whether the tree decoder architecture proposed here has advantages over previously proposed tree decoders, experimental comparison with alternative architectures will be helpful. Significance Tree to tree transduction is an important problem, and this paper is the first to propose successful neural models that explicitly models the tree structure in both the input and output. It is further shown that the model obtains a strong improvement over previous models in the practical task of program translation. While there is still a lot of work to do done in this area, this paper makes a strong contribution and I would like to see it accepted.

Reviewer 3



The paper presents a neural architecture for programming language translation based on tree-to-tree LSTM model and attention. The architecture is overall well described, there are sufficient details to replicate the results in the paper. The ideas are clearly described and the problem is an interesting area explored in the software engineering community. The approach with the attention on trees is original, but otherwise the tree-to-tree model is straightforward. Pros: The problem is interesting, with several competing works. Extensive evaluation and state-of-the-art performance. I particularly like that evaluation is broad and handles a range of programming language pairs. The metric is also realistic -- program accuracy is the realistic metric for such kind of app. Cons: I expect a bit more depth for NIPS. In particular it is not clear what is the "expressibility" of the proposed architecture. For example, for CoffeeScript to JavaScript one would expect that accuracy of 100% should be achievable. Is there a particular construct that cannot be captured by the model? Probably some form of attention that locates a source subtree based on a current position may demonstrate that the approach is at least comparable in expressibility to recursively defined tree-to-tree transformation functions. Minor: the paper discusses token accuracy, but the metric is only used in appendix. Author remarks: Thanks for the explanation. The cases in which the model fails to translate CS to JS likely points out that the model lacks expressibility, but nevertheless works well on many examples. It would be good if the authors expand the appendix in the final version to discuss the cases where the model fails, I am sure there are more cases than just temporary variables.